# Effect of Folic Acid Treatment for Patients with Traumatic Brain Injury (TBI)-Related Hospital Acquired Pneumonia (HAP): A Retrospective Cohort Study

**DOI:** 10.3390/jcm11247403

**Published:** 2022-12-14

**Authors:** Hao Wu, Xin Geng, Chenan Liu, Augustine K. Ballah, Feixiang Li, Tangrui Han, Shuai Gao, Chunhong Wang, Hongming Ji, Xiaoqi Nie, Gang Cheng, Xiangyu Wang, Rui Cheng, Yonghong Wang

**Affiliations:** 1Department of Neurosurgery, Shanxi Bethune Hospital, Shanxi Academy of Medical Sciences, Tongji Shanxi Hospital, Third Hospital of Shanxi Medical University, Taiyuan 030032, China; 2Department of Neurosurgery, The First Affiliated Hospital of Jinan University, Guangzhou 510630, China; 3The First Clinical Medical College of Jinan University, Guangzhou 510630, China; 4Key Laboratory of Cancer FSMP for State Market Regulation, Beijing 100038, China; 5Beijing Shijitan Hospital, Capital Medical University, Beijing 100038, China; 6Department of Neurosurgery, The First Affiliated Hospital of Shanxi University, Taiyuan 030001, China; 7Department of Neurosurgery, Shanxi Provincial People’s Hospital, Taiyuan 030012, China; 8Shanxi Provincial Key Laboratory of Intelligent, Big Data and Digital Neurosurgery, Taiyuan 030012, China

**Keywords:** Hospital Acquired Pneumonia, traumatic brain injury, folic acid

## Abstract

Hospital Acquired Pneumonia (HAP) is one of the most common complications and late causes of death in TBI patients. Targeted prevention and treatment of HAP are of great significance for improving the prognosis of TBI patients. In the previous clinical observation, we found that folic acid treatment for TBI patients has a good effect on preventing and treating HAP. We conducted this retrospective cohort study to demonstrate what we observed by selecting 293 TBI patients from two medical centers and analyzing their hospitalization data. The result showed that the incidence of HAP was significantly lower in TBI patients who received folic acid treatment (44.1% vs. 63.0%, *p* = 0.012). Multivariate logistic regression analysis showed that folic acid treatment was an independent protective factor for the occurrence of HAP in TBI patients (OR = 0.418, *p* = 0.031), especially in high-risk groups of HAP, such as the old (OR: 1.356 vs. 2.889), ICU (OR: 1.775 vs. 5.996) and severe TBI (OR: 0.975 vs. 5.424) patients. At the same time, cohort studies of HAP patients showed that folic acid also had a good effect on delaying the progression of HAP, such as reducing the chance of tracheotomy (26.1% vs. 50.8%, *p* = 0.041), and reduced the length of hospital stay (15 d vs. 19 d, *p* = 0.029) and ICU stay (5 d vs. 8 d, *p* = 0.046). Therefore, we believe that folic acid treatment in TBI patients has the potential for preventing and treating HAP, and it is worthy of further clinical research.

## 1. Introduction

Traumatic brain injury (TBI) refers to brain damage caused by trauma, car accidents, sports injuries, etc., and is a major global health problem. More than tens of millions of people worldwide are affected by TBI each year, resulting in an economic burden of hundreds of billions of dollars [1]. Deaths caused by TBI can be divided into early and late stages. Early deaths are caused by brain tissue damage, secondary hemorrhage, and edema. Late deaths are mostly caused by infection and multiple organ failure [2]. According to the National Institute of Disability and Rehabilitation in the United States, even if patients with moderate to severe TBI receive standardized rehabilitation treatment, the mortality rate is still 2.2 times that of the general population. Furthermore, they lost 6.6 years of life expectancy and were 44 times more likely to die from pneumonia than the general population [3]. Infection after TBI leads to a longer hospital and ICU stay, increased ventilator use, and increased risk of organ failure [4].

Studies have shown that hospital-acquired pneumonia (HAP) is an independent predictor of poor prognosis 5 years after discharge in patients with severe TBI [5]. Severe pneumonia in patients with TBI, especially acute respiratory distress syndrome (ARDS), will directly lead to death [6]. Therefore, the prevention and treatment of TBI-related HAP become the key to improving the prognosis of TBI patients. The currently recognized cause of TBI-related HAP is neurological impairment. TBI patients cannot excrete respiratory secretions, resulting in aspiration and infection. Therefore, clinical treatment for HAP patients mainly focuses on strengthening patient care, assisting in the discharge of oral secretions and prophylactic use of antibiotics. However, even with the early use of tracheostomy to enhance the excretion of airway secretions, the infection status and prognosis of the patients do not appear to be significantly improved [7]. Prophylactic antibiotics also do not appear to reduce the incidence of TBI-related HAP and even lead to the risk of multidrug-resistant bacterial infections [8]. Other drugs, such as progesterone and TNF-α inhibitors, appear to be effective in treating neuronal damage caused by TBI but do not improve patient outcomes and reduce complications [9].

Studies have shown that various micronutrient deficiencies, especially folic acid deficiency, are associated with poor prognosis in patients with pneumonia, especially in patients with COVID-19 [10,11]. However, studies are lacking in proving whether micronutrient supplementation is effective in pneumonia treatment. In long-term clinical work, we found that adjuvant folic acid treatment for patients with TBI-related HAP can significantly protect the respiratory mucosa and effectively control the progression of HAP (as shown in Figure 1). In this study, we collected the clinical data of 293 TBI patients and the application of folic acid treatment to explore whether folic acid treatment can be used as an effective treatment factor to prevent TBI-related HAP.

## 2. Research Objects and Methods

### 2.1. Research Objects and Inclusion/Exclusion Criteria

This retrospective study was conducted on TBI patients treated in Shanxi Bethune Hospital and Shanxi Provincial People’s Hospital from February 2020 to February 2022.

Inclusion criteria: (1) a clear history of craniocerebral trauma on admission, including car accident, fall, fight, etc.; (2) age > 18 years old; (3) cranial imaging examination showed brain contusion, skull fracture, intracranial hematoma and other post-traumatic changes; (4) all patients’ treatments followed treatment norms and guidelines, and no obvious medical accidents occurred during treatment.

Exclusion criteria: (1) patients or their family members refused to cooperate with treatment or had disputes during admission, resulting in deviations in treatment information; (2) combined with severe trauma to other organs or other serious diseases; (3) the condition was extremely critical at the time of admission, resulting in poor prognosis, such as brain herniation, respiratory and cardiac arrest, etc.; (4) death after rapid aggravation of the condition within 3 days after admission; (5) lack of important information in inpatient medical records, and subsequent analysis cannot be completed; (6) severe pulmonary disease was present at the time of admission.

The treatment and care received by all patients were evaluated in detail while reviewing the medical records of all patients. Only patients who were treated and cared in strict accordance with the guidelines for the treatment of TBI were included in the final study. Therefore, all patients included in the final study received standard, reasonable, and similar treatment and care, including vital sign monitoring, medication, medical procedures, sputum suction care, limb care, turning, and patting. We strictly evaluated treatment and care protocols in order to minimize bias in treatment effect by different medical group.

### 2.2. The Diagnosis of HAP

The diagnosis of HAP was based on the diagnostic criteria for pneumonia jointly developed by the American Society of Infectious Diseases and the American Thoracic Society in 2007 (Clin Infect Dis, 2007, 44:S27-72) and “Guidelines for the Diagnosis and Treatment of Hospital-Acquired Pneumonia and Ventilator-Associated Pneumonia in Chinese Adults (2018 Edition)”. The diagnosis of HAP could be established using Chest X-ray or CT showing new or progressive infiltrative, consolidation or ground-glass opacity, plus 2 or more of the following 3 clinical symptoms: (1) fever, body temperature >38 °C; (2) purulent airway secretions; (3) peripheral blood white blood cell count >10 × 10^9^/L or <4 × 10^9^/L.

### 2.3. Evaluation of Folic Acid Treatment Status

All patients included in the folic acid treatment group must have taken oral folic acid treatment within 24 h after admission (injected through a gastric tube in patients who cannot eat) at a dose of not less than 0.4 mg per day. For non-pneumonic patients, folic acid treatment continued for at least 7 days. For patients with pneumonia, their folic acid treatment needed to continue until the end of the entire treatment course. All patients treated with folic acid who did not meet the above requirements were excluded from the study.

### 2.4. Clinical Data Collection

Neurosurgeons and postgraduates completed the data collection of all enrolled patients. The indicators included sex, age, BMI, smoking history, drinking history, admission to ICU, GCS score, brain surgery, endotracheal intubation time, the first white blood cell (WBC) test, bacterial test results, length of hospital stay, and whether infected with HAP. Among them, Age, BMI, GCS, endotracheal intubation time, and WBC were transformed into binary variables: age ≤ 45 were young people and age > 45 were older people; BMI < 24 were normal weight and BMI ≥ 24 was overweight; GCS ≤ 12 was light TBI and GCS > 12 were middle to serious TBI; endotracheal intubation time ≤ 24 h was early opening airway and endotracheal intubation time > 24 h was late opening airway; WBC ≤ 10 × 10^9^/L were normal and WBC > 10 × 10^9^/L were abnormal. All enrolled patients completed a head CT examination upon admission, and the neurosurgeon made a diagnosis and recorded the head injury with reference to the imaging report given by the radiologist.

The patient’s daily body temperature and blood cell test results were recorded after admission. At the same time, we also collected the length of hospital stay, the length of ICU stay, and the total cost of hospitalization for all patients. It should be noted that some necessary evaluation indicators, such as GCS scores, were subjective, and their accuracy in the medical records cannot be determined. Therefore, for all patients included in the study, we also arranged for neurosurgery specialists to evaluate the patient’s GCS score according to the patient’s medical records. The records with obvious deviations were corrected to ensure the accuracy of the results. The quality control of medical records was completed by three doctors with senior professional titles (C.W., X.Q., R.C).

### 2.5. Statistical Analysis

Continuous variables conforming to normal distribution were expressed as mean ± standard deviation (SD), and differences between groups were compared using *t*-test. Continuous variables that were not normal distribution were defined as median and interquartile range [M(P25, P75)], and nonparametric tests compared differences between groups. Categorical data were expressed as numbers (percentages) and compared using chi-square or Fisher’s exact test. At the same time, the Bonferroni correction was used for correction. Logistic regression analysis was used to assess the relationship between risk factors and HAP in TBI patients. Tolerance or Variance inflation factor (VIF) was used to assess multicollinearity between variables. When the Tolerance was less than 0.1 or the VIF was greater than 10, we consider that there was collinearity between factors. Factors with *p* < 0.05 in the multivariate logistic regression analysis were considered independent risk factors. In addition, to further evaluate the relationship between folic acid and HAP in different subgroups, we combined folic acid treatment with other independent risk factors for cross-grouping. We performed multivariate logistic regression analysis with mutual adjustment. All analyses were performed using R software (version 4.2.0, R Foundation for Statistical Computing). Two-sided *p* < 0.05 indicated a statistically significant difference.

## 3. Results

### 3.1. Demographic and Clinical Characteristics

We collected 293 TBI patients from two medical centers in 2020–2022. A total of 106 were excluded according to the exclusion criteria, and 187 were finally included in the final study (as shown in Figure 2). The mean age of these patients was 46.2 ± 12.2 years old. The majority of patients were male (78.6%). The number of patients admitted to ICU was 115 (61.5%), and the number of patients with moderate to severe traumatic brain injury (GCS ≤ 12) was 88 (47.1%). A total of 64 (34.2%) patients underwent emergency surgery, and 105 (56.1%) patients were diagnosed with HAP during hospitalization. A total of 63 (33.7%) patients underwent tracheal intubation, of which 49 (26.2%) patients completed tracheal intubation within 24 h after admission. All the patients diagnosed with HAP were sent for bacterial test at the first time. Finally, 41 (21.9%) samples had bacterial test results, including 13 (7.0%) cases of Gram-positive bacteria and 28 (15.0%) cases of Gram-negative bacteria. The length of hospital stay was slightly shorter in the folic acid treatment group than in the non-folic acid treatment group, but the difference was not statistically significant. The rest of the clinical information is shown in Table 1.

### 3.2. Cohort Study of TBI Patients with and without Folic Acid Treatment

TBI patients (n = 187) were divided into the folic acid treatment group (n = 68) and the non-folic acid treatment group (n = 119) according to whether the patients received standard folic acid treatment. The sex, age, and other basic data of the two groups of patients were compared. The results showed that, except for BMI, bacterial test results, and HAP, there was no statistical difference between the two groups (*p* > 0.05). Although there was no overall difference in endotracheal intubation time between the two groups, Bonferroni correction indicated that there was a difference between the non-intubated group and the group intubated within 24 h (*p* < 0.05). Similarly, Bonferroni correction indicated differences in bacterial test results between the two groups only in Gram-positive and uninfected patients (*p* < 0.05).

It is worth noting that the proportion of HAP in the folic acid treatment group (44.1%) was significantly lower than that in the non-folic acid treatment group (63.0%), and the difference was statistically significant (*p* = 0.012) (Table 1). This suggests that folic acid treatment may potentially reduce the risk of HAP in TBI patients.

### 3.3. Independent Risk Factors for TBI-Associated HAP

To further study the effect of folic acid treatment on the occurrence of HAP in TBI patients, we combined 12 other covariates (including Sex, Age, BMI, Smoke, Drinking, Hypertension, Diabetes, ICU admission, GCS, Surgery, Endotracheal intubation time ≤ 24, WBC) to perform logistic regression analysis. It should be noted that airway exposure, namely, tracheal intubation, is also a factor affecting HAP. However, only early tracheal intubation can affect HAP, and late tracheal intubation may be the result of HAP aggravation. In view of the fact that most TBI patients in this study were intubated within 24 h after admission, and the remaining patients were intubated after HAP occurred 24 h after admission and their condition worsened, so Endotracheal intubation time ≤ 24 was considered as one of the covariates.

First, the diagnosis indicated no collinearity among the factors (VIF < 10). After adjustment by other covariates, folic acid treatment remained a protective factor for HAP in TBI patients (*p* = 0.031). In addition, Age > 45 (*p* = 0.001), ICU patient (*p* = 0.012), and GCS ≤ 12 (*p* = 0.041) were also independent risk factors for HAP in TBI patients (as shown in Figure 3).

### 3.4. The Role of Folic acid Treatment in Different TBI Subgroups

We regrouped folic acid treatment with 3 other independent risk factors and performed multivariate logistic regression analysis with mutual adjustment. 

The results showed that without folic acid treatment, the risk of developing HAP in every high-risk TBI subgroup would significantly increase (non-folic acid treatment + Age > 45: OR: 2.889 [CI: 1.161, 7.186], *p* = 0.023; non-folic acid treatment + GCS ≤ 12: OR: 5.424 [CI: 1.931, 15.233], *p* = 0.001; non-folic acid treatment+ICU patient: OR: 5996 [2.212, 16.251], *p* < 0.001). After using folic acid, the risk of HAP caused by these risk factors was significantly reduced or even completely offset (folic acid treatment + Age > 45: OR: 1.356 [CI:0.505,3.640], *p* = 0.545; folic acid treatment + GCS ≤ 12: OR: 0.975 [CI: 0.363, 2620], *p* = 0.950; folic acid treatment + ICU patient: OR: 5.996 [0.582, 5.414], *p* = 0.313). At the same time, we found that although some of the results were not statistically significant, the folic acid treatment also reduced the risk of HAP in low-risk TBI subgroups (folic acid treatment + Ag ≤ 45: OR: 0.244 [CI: 0.067, 0.895], *p* = 0.023; folic acid treatment + GCS > 12: OR: 0.925 [CI: 0.321, 2.666], *p* = 0.885; folic acid treatment + Ordinary patient: OR: 0.640 [0.201, 2.035], *p* = 0.449). (Figure 4).

### 3.5. Therapeutic Effect of Folic Acid on TBI-Related HAP

We selected ICU patients with HAP after TBI as the research target. We divided TBI patients into the folic acid treatment group (n = 23) and the non-folic acid treatment group (n = 63) according to whether the patients received standard folic acid treatment. 

The reason for including only ICU patients was a huge difference in hospitalization costs between ICU patients and ordinary patients. At the same time, the number of ordinary patients undergoing tracheal intubation and tracheostomy was 0, so it was not suitable for inclusion in the study. Tracheotomy was the main outcome index we included because the purpose of tracheotomy was generally for better sputum drainage, which was always used in patients with long-term endotracheal intubation or severe pneumonia. This invasive operation was harmful to patients, so that it can be seen as a manifestation of the rapid progression of HAP. Other outcomes were the length of hospital stay (day), length of ICU stay (day), and hospital costs (yuan).

A total of 86 ICU patients developed HAP after TBI, and 38 (44.2%) patients underwent tracheotomy after admission. The median length of hospital stay was 18 days, the median length of ICU stay was 7 days, and the median hospital cost was 104,921 yuan. The results showed that 6 (26.1%) of the patients in the folic acid treatment group underwent tracheostomy, which was significantly lower than the non-folic acid treatment group (32 (50.8%), *p* = 0.041). The median time after admission to tracheotomy was 120 h in the folic acid treatment group, which was longer than the non-folic acid treatment group (96 (52, 122), *p* = 0.041), although the difference was not statistically significant. Meanwhile, using folic acid treatment, patients’ total hospital stay (folic acid treatment: 15 (12, 17); non-folic acid treatment: 19 (12, 31); *p* = 0.029) and ICU stay (folic acid treatment: 5 (2, 10); non-folic acid treatment: 8 (5, 14); *p* = 0.046) were also significantly shortened. The rest of the outcome information is shown in Table 2.

Although folic acid treatment shortened the length of hospital stay, there was no significant difference in the total hospital cost between the two groups. We reviewed the patient’s medical order list and found that this may be related to the treatment regimen after the patient was transferred from the ICU. Patients in the folic acid treatment group recovered faster and chose to start rehabilitation treatment directly, thus increasing the treatment cost, while patients in the non-folic acid treatment group recovered slower and chose to transfer to the rehabilitation hospital for further treatment after the condition was stable, thus reducing the treatment cost. Of course, it would be helpful to analyze the components of treatment costs such as drugs, antibiotics, and nursing care, but unfortunately, we only got the total cost of patients’ stay from the health system.

## 4. Discussion

Folic acid is a B vitamin, known as vitamins B9, M, Bc, etc. It is a water-soluble vitamin whose chemical formula is C_19_H_19_N_7_O_6_ [12]. The human body does not produce folic acid. Folic acid in the human body mainly comes from food absorbed from the duodenum or colon [12]. After folic acid is absorbed, it converts into dihydrofolate (DHF), tetrahydrofolate (THF), and 5-methyl-THF in intestinal epithelial cells or the liver and enters the blood circulation to reach various tissue cells. Folic acid is an important carrier of the one-carbon pathway [13]. As a key factor in maintaining normal cell activities, folic acid is mainly involved in the methylation of various molecules and synthesizing purine, pyrimidine, choline, and other substances [13]. Folic acid deficiency can lead to fetal neural tube defects and anemia and can also lead to hyperhomocysteinemia, which in turn increases the risk of cerebrovascular disease [14,15]. Studies have also shown that folic acid deficiency and hyperhomocysteinemia can lead to cognitive impairment and even Alzheimer’s disease [15]. Therefore, folic acid is mainly used to prevent fetal developmental malformations and cerebrovascular diseases. 

Some studies have shown various micronutrient deficiencies in patients who suffer from pneumonia, especially COVID-19, and folic acid deficiency is associated with poor prognosis in patients with pneumonia [10,11]. However, there are no relevant studies on vitamin supplementation for the prevention and treatment of pneumonia, and no guidelines or consensuses suggest that HAP patients need additional folic acid supplementation.

HAP is the most common complication of TBI patients. This study showed that the incidence of HAP in TBI patients was as high as 56.1%, which is also consistent with the results reported by many other studies. Through experience and observation, we have found that folic acid treatment in TBI patients has a good effect on the prevention and treatment of HAP (as shown in Figure 1). So why is this effect not first discovered in the ordinary HAP patient population? This abnormality in the discovery sequence of the therapy may indicate that folic acid may not play a simple therapeutic role in HAP. At the same time, there are differences between TBI-related HAP and ordinary HAP, which may be related to folic acid deficiency. The differences are mainly in two aspects.

1. Patients with TBI-related HAP are more likely to have nutritional risk. TBI increases the risk of HAP secondary to neuronal damage, including impaired consciousness, dysphagia, vomiting, impaired cough reflex, prolonged bed rest, etc... This risk increases with the severity of TBI as our study showed that moderate to severe TBI (GCS score < 12) was an independent risk factor for HAP (*p* = 0.042). This view is the main reason for HAP in TBI patients. It is worth noting that the disturbance of consciousness caused by TBI also affects the nutrient intake of patients. Neurosurgeons often choose parenteral nutrition in the acute stage of TBI to give patients sufficient energy.

The nutrients are mainly sugar, fat, and protein, but they often ignore the supplement of other trace elements such as vitamins, resulting in nutritional risk. It can be proved by the patient’s medical records and wills in our study. Ordinary HAP patients will not have severe neuronal damage and disturbance of consciousness. Therefore, regular HAP patients can usually expectorate and eat independently, and their nutritional risk of various trace elements required is lower than that of TBI patients. Their vitamin deficiency may not be obvious [10].

2. Patients with TBI-related HAP may have more severe infections. TBI can directly cause damage to the respiratory and immune systems, which is the conclusion of research in recent years. After TBI, due to abnormal secretion of adrenal hormones, activation of damage-associated molecular patterns (DAMPs), and release of cytokines, diseases such as acute lung injury, pulmonary edema and even ARDS may occur in the lungs (as shown in Figure 1 severe damage) [2].

Inflammatory factors will also pass through the damaged blood–brain barrier and affect the cranial immune cells, further increasing the permeability of the blood–brain barrier and aggravating the central nervous system [16]. At the same time, TBI reduces the activity of neutrophils, NK cells, and other immune cells through multiple pathways [4,16]. It reduces the production of pro-inflammatory factors such as interleukin (IL)-1β and tumor necrosis factor (TNF)-α and increases the production of the anti-inflammatory cytokine IL-10, which results in immunosuppression and worsening of infection [4]. Therefore, TBI-related HAP is often more severe than ordinary HAP. Furthermore, the high metabolic state caused by severe infection will increase the consumption of nutrients, which creates a high demand for folic acid to maintain cell proliferation and organ repair. If folic acid is not enough, then the normal metabolism of cells will be affected, which blocks the proliferation of lymphocytes and the repair of tracheal epithelial cells, further aggravating HAP.

Combining the above two points, patients with TBI-related HAP have a higher risk of developing folic acid deficiency. Meanwhile, studies have shown that regulatory T cells (Treg) express high levels of folic acid receptor (folate receptor 4 [FR4]) [17]. Folic acid can promote Treg proliferation by binding to FR4, which contributes to the maintenance of immunologic homeostasis [17,18,19,20]. Therefore, the lack of folic acid will further lead to immune cell dysfunction on the basis of DAMPs, resulting in immunologic homeostasis disturbance and infection aggravation.

This study found that folic acid treatment in TBI patients could effectively prevent the occurrence of HAP. This effect was more pronounced in patients with old age, severe TBI, and ICU admission. These groups of TBI patients were prone to a lack of normal enteral nutrition supply and folic acid deficiency, which confirms our conjecture from the side. At the same time, in the TBI patients with HAP, the probability of needing tracheostomy was significantly decreased after folic acid supplementation (50.8% to 26.1%, *p* = 0.041), and the lengths of ICU stay and hospital stay were shortened, indicating that folic acid also had an effective therapeutic effect on HAP. Based on the above theories, folic acid does not directly affect HAP like antibiotics. Still, the special state after TBI may make patients stay in a pathological state of lack of folic acid, then leading to the occurrence and aggravation of HAP. Of course, our theory is a conjecture based on the results of this study, and further prospective studies are needed to confirm the true folic acid status of TBI patients.

This study has the following shortcomings: (1) This study is a retrospective study, and there are uncontrollable confounding factors, such as the dose of folic acid treatment, the pathogenic flora of HAP, and the type and dose of antibiotics used for treatment. (2) The number of cases is too small; especially after the subgroup analysis, the further reduction of the number of cases causes the risk of statistical analysis. It may make some conclusions unreliable, such as the 95% CI range in the multivariate logistic regression analysis being too large. (3) The retrospective study data lack elements to prove the argument of this paper, such as the test results of folic acid and lymphocyte subsets in the blood. (4) There is a lack of experimental research on the effect of folic acid on lymphocytes and tracheal epithelial cells after HAP.

In the future, we will further carry out prospective, multicenter clinical studies; expand the sample size; control confounding factors; and obtain more complete clinical data to prove the thesis of this paper. At the same time, experimental research will be carried out to determine the specific mechanism of folic acid in treating HAP.

## 5. Conclusions

In conclusion, this study proves that folic acid treatment is an independent influencing factor of HAP in TBI patients. Folic acid treatment can effectively prevent the occurrence of HAP in TBI patients, especially in elderly and severe TBI patients. Folic acid can also delay the progression of HAP and effectively shorten the hospitalization time of patients, which is worthy of continued exploration of clinical treatment.

## Figures and Tables

**Figure 1 jcm-11-07403-f001:**
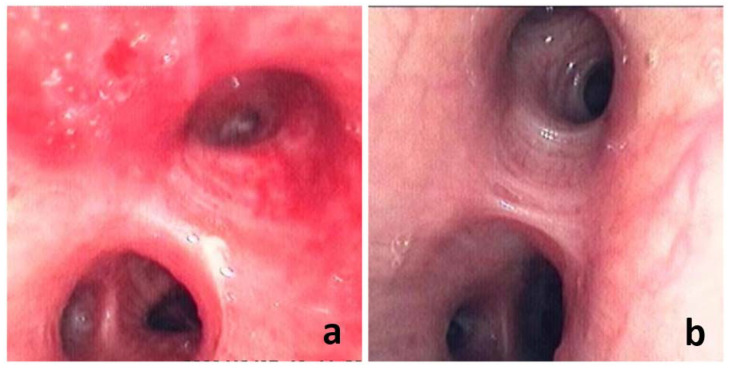
Bronchoscopic images of two patients with TBI. (**a**) 71-year-old male, GCS score was 5, was diagnosed with HAP on the third day and without folic acid treatment after admission. The bronchoscopic image was taken 19 days after the patient’s admission. The image showed that the patient had significant damage to the tracheal mucosa. (**b**) 69-year-old male, GCS score was 5, was diagnosed with HAP on the third day and with folic acid treatment after admission. The bronchoscopic image was taken 16 days after the patient’s admission. The image showed that the patient’s tracheal mucosa was intact.

**Figure 2 jcm-11-07403-f002:**
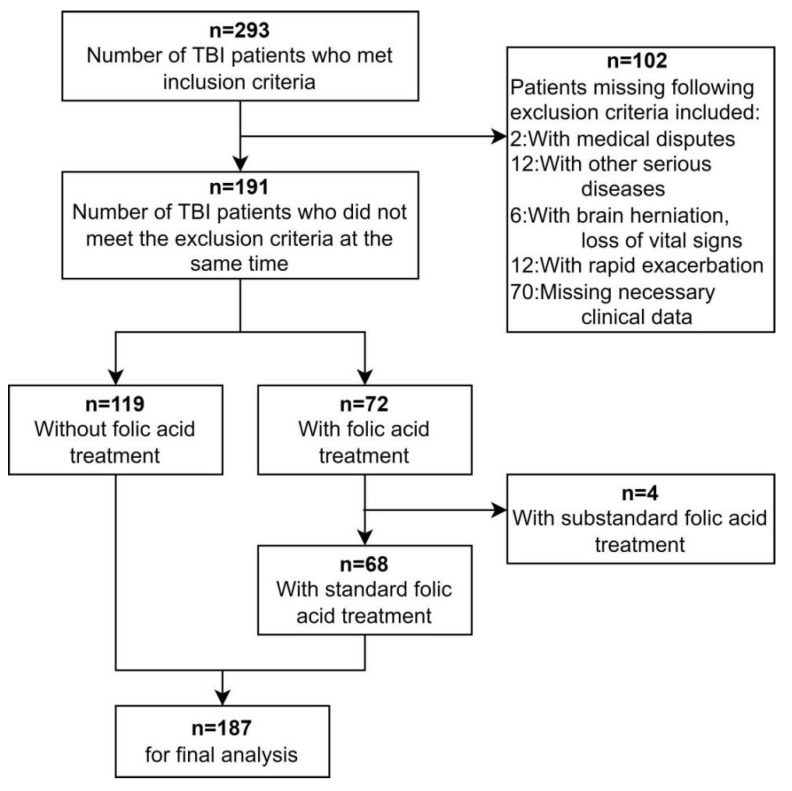
Flowchart for patient inclusion.

**Figure 3 jcm-11-07403-f003:**
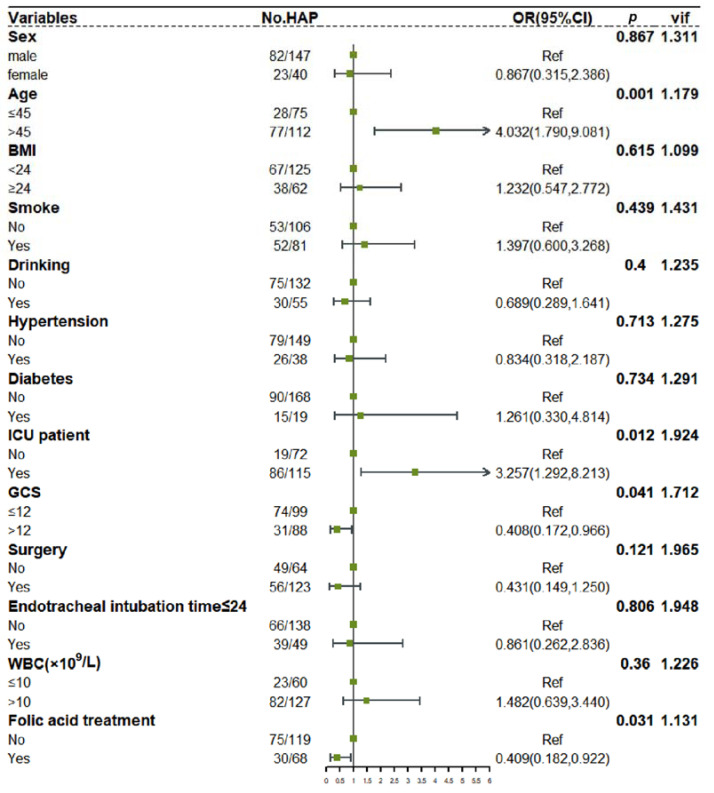
Multivariate logistic regression results and forest plot of the risk of HAP in TBI patients. Age > 45 (*p* = 0.001), ICU patient (*p* = 0.012), GCS ≤ 12 (*p* = 0.041) and folic acid treatment (*p* = 0.031) were independent risk factors for HAP in TBI patients.

**Figure 4 jcm-11-07403-f004:**
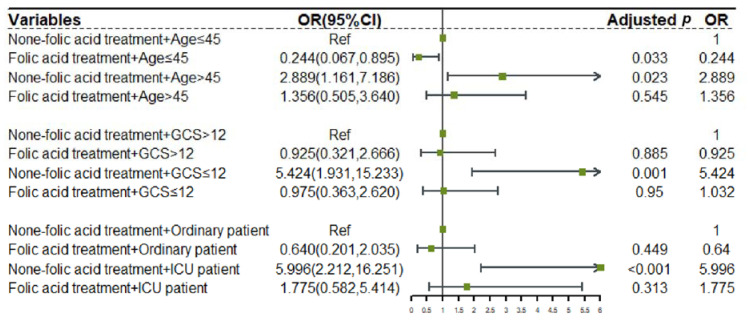
Forest plot of the effect of folic acid treatment. We combined folic acid treatment with other independent risk factors (Age, GCS and ICU patient) for cross-grouping and performed multivariate logistic regression analysis with mutual adjustment.

**Table 1 jcm-11-07403-t001:** Clinical information of overall TBI patients, folic acid treatment and non-folic acid treatment group patients.

		Overall TBI Patients (n = 187)	Subgroup According to Treatment			
			Folic Acid Treatment (n = 68)	Non-Folic Acid Treatment (n = 119)	χ^2^/Fisher/Z	*p*
Sex n (%)	male	147 (78.6)	56 (82.4)	91 (76.5)	0.890	0.345
	female	40 (21.4)	12 (17.6)	28 (23.5)		
Age(years) n (%)	≤45	75 (40.1)	29 (42.6)	46 (38.7)	0.287	0.592
	>45	112 (59.9)	39 (57.4)	73 (61.3)		
BMI n (%)	≤24	125 (66.8)	54 (79.4)	71 (59.7)	7.614	0.006
	>24	62 (33.2)	14 (20.6)	48 (40.3)		
Smoke n (%)	no	106 (56.7)	39 (57.4)	67 (56.3)	0.019	0.889
	yes	81 (43.3)	29 (42.6)	52 (43.7)		
Drinking n (%)	no	132 (70.6)	50 (73.5)	82 (68.9)	0.445	0.505
	yes	55 (29.4)	18 (26.5)	37 (31.1)		
Hypertension	no	149 (79.7)	58 (85.3)	91 (76.5)	2.081	0.149
N (%)	yes	38 (20.3)	10 (14.7)	28 (23.5)		
Diabetes n (%)	no	168 (89.8)	64 (94.1)	104 (87.4)	2.142	0.143
	yes	19 (10.2)	4 (5.9)	15 (12.6)		
ICU patient	no	72 (38.5)	30 (44.1)	42 (35.3)	1.423	0.233
N (%)	yes	115 (61.5)	38 (55.9)	77 (67.7)		
GCS n (%)	>13	99 (52.9)	37 (54.4)	62 (52.1)	0.093	0.761
	≤12	88 (47.1)	31 (45.6)	57 (47.9)		
Surgery n (%)	no	123 (65.8)	48 (70.6)	75 (63.0)	1.100	0.294
	yes	64 (34.2)	20 (29.4)	44 (37.0)		
Endotracheal	no	124 (66.3)	52 (76.5)	72 (60.5)	5.766	0.055 ^a^
intubation time	≤24	49 (26.2)	11 (16.2)	38 (31.9)		
(h) n (%)	>24	14 (7.5)	5 (7.3)	9 (7.6)		
WBC(×10^9^/L)	≤10	60 (32.1)	24 (35.3)	36 (30.3)	0.505	0.477
N (%)	>10	127 (67.9)	44 (64.7)	83 (69.7)		
HAP n (%)	no	82 (43.9)	38 (55.9)	44 (37.0)	6.238	0.012
	yes	105 (56.1)	30 (44.1)	75 (63.0)		
Bacterial test	Gram+	13 (7.0)	1 (1.5)	12 (10.1)	10.275	0.015 ^b^
results n (%)	Gram−	28 (15.0)	11 (16.2)	17 (14.3)		
	unknown	64 (34.2)	18 (26.5)	46 (38.7)		
	none	82 (44.8)	38 (55.9)	44 (37.0)		
Length of hospital stay (day)		13 (9, 19)	12.5 (9, 16)	13 (9, 22.5)	−1.480	0.139

Note. ^a^. As for Endotracheal intubation time factor, the “no” group was the patients without endotracheal intubation, the “≤24” group was the patients with endotracheal intubation within 24 h of admission, and the “>24” group was the patients with endotracheal intubation 24 h after admission. Chi-square test indicated that there were no differences in tracheal intubation time between the two treatment groups. But Bonferroni correction indicated that at the level of α = 0.05, there was a difference in “no” and “≤24” groups. ^b^. As for bacterial test results factor, the “Gram+” group was the patients infected with Gram-positive bacterium and the “Gram−” group was the patients infected with Gram-negative bacterium. The “unknown” group was the HAP patients without certain bacterial test results, and the “none” group was uninfected patients. Fisher’s exact test indicated that there were significant differences in bacterial test results between the two treatment groups. But Bonferroni correction indicated that at the level of α = 0.05, There was a difference in groups “Gram+” and “none” but no difference in groups “Gram−” and “unknown”.

**Table 2 jcm-11-07403-t002:** Outcomes of overall ICU patients with HAP after TBI, folic acid treatment, and non-folic acid treatment group patients.

		Overall HAP Patients in ICU (n = 86)	Folic Acid Treatment (n = 23)	Non-Folic Acid Treatment (n = 63)	χ^2^/Z	*p*
Tracheotomy	No	48 (55.8)	17 (73.9)	31 (49.2)	4.170	0.041
n (%)	Yes	38 (44.2)	6 (26.1)	32 (50.8)		
Tracheotomy time (h)		96 (52, 126)	120 (96, 223)	96 (52, 122)	−1.606	0.108
Length of hospital stay (day)		18 (12, 29)	15 (12, 17)	19 (12, 31)	−2.183	0.029
Length of ICU stay (day)		7 (4, 13)	5 (2, 10)	8 (5, 14)	−2.000	0.046
Hospital costs (yuan)		104,921 (51,528, 156,318)	107,975 (33,913, 132,797)	104,143 (55,227, 178,309)	−1.381	0.167

## Data Availability

The data after desensitization treatment and presented in this study are available on request from the first author (X.G.).

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
