# Peer review of "Effect of Folic Acid Treatment for Patients with Traumatic Brain Injury (TBI)-Related Hospital Acquired Pneumonia (HAP): A Retrospective Cohort Study"

_jcm, 2022, doi:10.3390/jcm11247403_

Round 1
Reviewer 1 Report
This is very interesting topic.
I have the following comments:
There is an important difference between the number of patients enrolled in the folic acid group compared with the control group. In the control group, there are almost twice as many patients. Also, this kind of study could be very interesting if it included some details: understanding which pathogens were involved in nosocomial pneumonias, if folic acid has a greater effect against Gram+ or Gram- patients. More specifics would be needed regarding the number of patients intubated, how long they had been hospitalized, at what stage they were intubated if they were and at what stage they were tracheostomized if they were. Furthermore, it would be necessary to define what devices were used for the management of the airway aspiration toilet, if the nurses have protocols for the management of these patients, and everything about infection vehicles in nosocomial infections. Finally, this study may be more interesting and may make more contribution to the literature if these adjustments were made using graphs or tables and if the number of patients divided into subgroups was more equitable.
Author Response
- There is an important difference between the number of patients enrolled in the folic acid group compared with the control group. In the control group, there are almost twice as many patients.
This is a preliminary retrospective study. The role of folic acid therapy in the treatment of TBI-related HAP was first discovered and attempted to treat by the treatment group of the first author. At the time of case collection and analysis, the patients in the folic acid treatment group were also almost all patients in this treatment group. Thus, the number of patients in the folate-treated group would have been smaller than in the control group. Of course, quality control of medical records was performed to ensure that all enrolled patients received standard care and that there were no substantial differences in treatment between different groups. Cases with treatment errors were excluded. As for the difference in the number of patients between the treatment group and the control group as you mentioned, after consulting a professor majoring in statistics from Shanxi Medical University, we got the answer that the number of patients in the two groups should be as equal as possible. But for some rare diseases or new treatment methods, the number of patients may be small, and the number of patients in the control group may be more or even double. For example, some papers published in JCM recently also use 1:2 ratio in the experimental group and control group of patients, detail please refer to: https://doi.org/10.3390/jcm11195855. At the same time, thank you very much for your suggestion. We hope to include more cases in the following study and try to balance the number of patients in the two groups, so as to make the study results more accurate.
- Also, this kind of study could be very interesting if it included some details: understanding which pathogens were involved in nosocomial pneumonias, if folic acid has a greater effect against Gram+ or Gram- patients.
The centers that supported this study had inadequate bacteria test for patients with TBI, resulting in a lack of bacteria information. Of 105 patients with HAP, only 40 had definitive microbiologic findings. The reasons for this consideration were as follows: 1. The neurosurgeon chose to administer empirical intravenous antibiotic therapy first after the patient developed symptoms of pneumonia, and then send a sputum test. We consider that early antibiotic treatment may have affected the microbiologic test positivity. 2. If the first test result is negative, some neurosurgeon in charge may not choose to send the bacteria culture again, but continue effective treatment according to the patient's symptoms and the nature of the sputum. So this is very important to us, and we are aware of the experimental shortcomings in the detection of pathogenic microorganisms, so we have selected better microbiological detection protocols in our ongoing clinical prospective studies to ensure that the pathogen species are identified in every patient with pneumonia. As you said, this will allow us to find out if there are differences in folic acid treatment in different types of microbial infections. In summary, it is unfortunate that we are only able to provide baseline data on the microbiologic species in Table 1, but the number of meaningful results did not support the analysis in this study. Of course, we will analyze your suggestion in detail in the follow-up study.
- More specifics would be needed regarding the number of patients intubated, how long they had been hospitalized, at what stage they were intubated if they were and at what stage they were tracheostomized if they were.
This is a critical piece of advice. According to previous studies, airway opening plays an important role in the development of pneumonia, but we ignored this key factor in the first analysis. Therefore, we reorganized some of the authors to conduct a brief review of all the patients' medical records again, collecting all the patients' tracheal intubation status and time and listing it in Table 1. The results showed that almost all endotracheal intubations were completed within 24 hours after admission. The reason for tracheal intubation in these patients was the aggravation of disturbance of consciousness, while the tracheal intubation after 24 hours was mostly due to the aggravation of pneumonia. Therefore, we included tracheal intubation within 24 hours of admission as an influencing factor in the multivariate regression analysis. However, the results showed that endotracheal intubation did not seem to affect the happening of HAP in this study. In addition, the length of hospital stay and the timing of tracheotomy for all patients have been included in Tables 1 and 2. The results showed a slight reduction in the length of hospital stay and a slight delay in the timing of tracheotomy in patients with TBI in the ICU, although these differences were not statistically significant.
- Furthermore, it would be necessary to define what devices were used for the management of the airway aspiration toilet, if the nurses have protocols for the management of these patients, and everything about infection vehicles in nosocomial infections.
The treatment and care received by all patients were evaluated in detail while reviewing the medical records of all patients. Only patients who were treated and cared in strict accordance with the guidelines for the treatment of TBI were included in the final study. Patients who did not comply were excluded on the grounds of "Missing necessary clinical data" (see Flowchart for patient inclusion). Therefore, the 187 TBI patients in the final statistical analysis received standard, reasonable, and similar treatment and care, including vital sign monitoring, medication, medical procedures, sputum suction care, limb care, turning and patting. At the same time, the sputum suction equipment used by patients was the same model purchased by the Shanxi Provincial Health Department. All nurses also had qualification certificates, and the quality of ICU was also controlled by the hospital. Therefore, after consultation, the authors in this article concluded that there were no differences in general medical support and nursing support that were sufficient to affect patient outcomes. Of course, these factors are to be reflected in the article, and thank you very much for your question. We have described these factors you proposed in a separate paragraph.
- Finally, this study may be more interesting and may make more contribution to the literature if these adjustments were made using graphs or tables and if the number of patients divided into subgroups was more equitable.
All of our modifications are reflected in the table in this article, including the length of hospital stay, the number and timing of tracheal intubation, the results of bacterial test, and the timing of tracheotomy. The problem of unequal numbers of patients in the subgroup was also described in the discussion. This study was a retrospective study, and the number of patients with TBI who met the requirements and received folic acid treatment was small. When subgroup analysis was performed, the number of some subgroups was too small, which was indeed a defect of this paper. In the future prospective study, we will collect a sufficient number of TBI patients to reduce the statistical risk caused by too few cases or uneven distribution, and provide more evidence for the clinical application of folic acid treatment.
Reviewer 2 Report
This is an interesting retrospective cohort study of patients with traumatic brain injury (TBI) who received folic acid treatment to determine the effects on hospital acquired pneumonia (HAP). The abstract is complete but would benefit from actual p values note with the significance. It would also be good to include the details of the folic acid treatment used.
The introduction has strong support for why reduction of HAP is important in patients with TBI. There appears to be an adequate review of current literature.
The inclusion and exclusion are described in adequate detail in the methods and appear appropriate for the study. The diagnosis of HAP was based on international guidelines and is well described and referenced. The description of the treatment includes the minimum amount of folic acid, route, and minimum days of treatment. Were there patients who received more than the minimum doses noted? That is not clear based on the description. Did doses differ among patients and if so, what determined the dosage?
The data collection is described with good detail related to data collected. It was noted that neurosurgeons and postgraduates collected the data. Was there any check to assure data were collected accurately or interpreted correctly? It would be helpful to know how costs of the hospitalization was determined as there are many ways to calculate that.
The statistical analyses are described well. There were many analyses completed so the chance for significant results is increased. There does not appear to have been a correction, such as a Bonferroni correction made in light of the many analyses.
The consort diagram is complete and easy to understand. The inclusion of bronchoscopy images in similar patients demonstrating the effects of folic acid were impressive.
The results are clearly presented. There is a good explanation for the covariates and the analysis. The figure is helpful and clear (Figure 3). The data are presented well. The only item that is interesting but not noted is that increase in costs for the folic acid treatment group. It is noted that the costs are not significantly different, but a brief statement might be helpful.
In the discussion, it is noted that parenteral nutrition does not include vitamins and trace elements while in many parts of the world, it would be included. This seems to be a potential recommendation in these findings. Is there any indication that some of these patients had nutritional risks prior to the TBI due to alcohol or chemical use?
The discussion notes the need for additional studies, however the conclusion notes that folic acid treatment can effectively prevent HAP in TBI patients. This seems to overstate the findings and contradict earlier statements.
Author Response
- This is an interesting retrospective cohort study of patients with traumatic brain injury (TBI) who received folic acid treatment to determine the effects on hospital acquired pneumonia (HAP). The abstract is complete but would benefit from actual p values note with the significance. It would also be good to include the details of the folic acid treatment used.
Thanks for your advice, we have supplemented the summary with a description of some key p values as well as details of the folic acid treatment used. At the same time, the details of the folic acid treatment used are described in more detail in the article.
- The inclusion and exclusion are described in adequate detail in the methods and appear appropriate for the study. The diagnosis of HAP was based on international guidelines and is well described and referenced. The description of the treatment includes the minimum amount of folic acid, route, and minimum days of treatment. Were there patients who received more than the minimum doses noted? That is not clear based on the description. Did doses differ among patients and if so, what determined the dosage?
This is a very good advice and points out a key shortcoming of this paper. As we said in the discussion, our study was retrospective, and the application of folic acid in the treatment of TBI patients was not recommended by the corresponding guidelines, which led to our blindness in the dosage of folic acid. According to the package insert and the dietary recommendations for U.S. residents (https://health.gov/; the original guideline appears to have changed location), we administered an oral folate dose of 0.4mg/ day to patients with TBI. However, for a small number of patients with larger body weight, longer disease duration, more severe pneumonia, or older patients with intestinal absorption difficulties, we may choose to use a moderate dose of folic acid (0.6-0.8mg/d). So, We describe in the article: "All patients included in the folic acid treatment group must have taken oral folic acid treatment within 24 hours after admission (injected through a gastric tube in patients who cannot eat) at a dose of not less than 0.4 mg per Day ". We will further determine the folic acid treatment regimen in a future prospective study.
- The data collection is described with good detail related to data collected. It was noted that neurosurgeons and postgraduates collected the data. Was there any check to assure data were collected accurately or interpreted correctly? It would be helpful to know how costs of the hospitalization was determined as there are many ways to calculate that.
This is a good advice, as we also found uneven quality of patient records, which can cause some unnecessary bias to appear. As mentioned in the "Clinical data collection" section of our article, "All enrolled patients completed a head CT examination upon admission, and the neurosurgeon made a diagnosis and recorded the head injury with reference to the imaging report given by the radiologist ", "It should be noted that some necessary evaluation indicators, such as GCS scores, were subjective, and their accuracy in the medical records cannot be determined. Therefore, for all patients included in the study, we also arranged for neurosurgery specialists to evaluate the patient's GCS score according to the patient's medical records. The records with obvious deviations were corrected to ensure the accuracy of the results ". The authors, C.W., X.Q., and R.C., who hold senior physician titles, were responsible for case quality control for all the patients in the study This point will be highlighted in revisions. As for hospitalization expenses, we have provided a supplementary description in the article (Results section). Due to ethical and medical insurance restrictions, we only obtained the total hospitalization expenses of patients. Of course, it is also helpful to analyze the components of treatment expenses such as drugs, antibiotics, and nursing care, which will be improved in the subsequent research.
- The statistical analyses are described well. There were many analyses completed so the chance for significant results is increased. There does not appear to have been a correction, such as a Bonferroni correction made in light of the many analyses.
This is an excellent advice, since statistical analyses of multicategory outcomes require tests such as Bonferroni correction. Therefore, we performed Bonferroni correction for the two factors of Endotracheal intubation time and Bacterial test results. This makes our statistical results richer and more accurate.
- The results are clearly presented. There is a good explanation for the covariates and the analysis. The figure is helpful and clear (Figure 3). The data are presented well. The only item that is interesting but not noted is that increase in costs for the folic acid treatment group. It is noted that the costs are not significantly different, but a brief statement might be helpful.
We used percentiles to describe hospitalization costs because they are not normally distributed, and the median hospitalization costs appeared to be slightly higher among patients who received folic acid than those who did not. In fact, on average, patients treated with folic acid had slightly lower hospital costs, although the difference was not statistically significant. Of course, we reviewed the patient's medical order list and found that this may be related to the treatment regimen after the patient was transferred from the ICU. Patients in the folic acid treatment group recovered faster and chose to start rehabilitation treatment directly, thus increasing the treatment cost. While patients in the non-folic acid treatment group recovered slower and chose to transfer to the rehabilitation hospital for further treatment after the condition was stable, thus reducing the treatment cost. This analysis is reflected in the revision.
- In the discussion, it is noted that parenteral nutrition does not include vitamins and trace elements while in many parts of the world, it would be included. This seems to be a potential recommendation in these findings. Is there any indication that some of these patients had nutritional risks prior to the TBI due to alcohol or chemical use?
In China, there is no medical guideline that considers vitamin supplementation, especially folic acid, as an essential element of parenteral nutrition for neurosurgical ICU patients. At the same time, almost all hospitals do not add vitamins to parenteral nutrition, but only ensure normal sugar, amino acids, and fat. Neurosurgeons tend to focus more on treating the primary disease, may ignore nutrition supply. The guidelines mostly recommend early enteral nutrition treatment and encourage patients to eat normal food as soon as possible, which has been proved to have a clear effect on the recovery of patients. Of course, this advice to eat as early as possible inadvertently increases the vitamin intake of patients and reduces the nutritional risk of TBI patients. This is also consistent with the results of our study that ordinary patients who can eat normally are a protective factor for HAP. We also hope that through our series of studies, Chinese neurosurgeons can understand the importance of folic acid and other vitamins supplementation for TBI patients. At the same time, none of the patients had a record of malnutrition caused by chemotherapy drugs or excessive alcohol consumption.
- The discussion notes the need for additional studies, however the conclusion notes that folic acid treatment can effectively prevent HAP in TBI patients. This seems to overstate the findings and contradict earlier statements.
Folic acid treatment had a preventive effect and a therapeutic effect on TBI-related HAP. These two conclusions were obtained from the multivariate logistic regression analysis and the analysis of TBI patients admitted to ICU. The conclusions are based on the cases collected in our current study and have certain limitations, and the conclusions in our original article are indeed too arbitrary, so they have been revised as suggested by you.
Round 2
Reviewer 1 Report
I would like to compliment with the Authors for the efforts provided in addressing my comments.
The manuscript is, in my opinion, suitable for publication and can be accepted in its current form.
Best regards